# Animal Training, Environmental Enrichment, and Animal Welfare: A History of Behavior Analysis in Zoos

Eduardo J. Fernandez [1,*] and Allison L. Martin [2]

1   School of Animal and Veterinary Science, The University of Adelaide, Adelaide, SA 5005, Australia
2   Department of Psychological Science, Kennesaw State University, Kennesaw, GA 30144, USA;
    allison.martin@kennesaw.edu
*   Correspondence: edjfern@gmail.com

**Abstract:** The modern zoo has been associated with two major behavioral welfare advances: (a) the use of training to increase voluntary husbandry care, and (b) the implementation of environmental enrichment to promote naturalistic behaviors. Both practices have their roots in behavior analysis, or the operant conditioning-centered, reward-based approach to behavioral psychology. Operant conditioning served as the foundation for the development of reinforcement-based training methods commonly used in zoos to make veterinary and husbandry procedures easier and safer for animals and their caregivers. Likewise, operant conditioning, with its focus on arranging environmental antecedents and consequences to change behavior, also provided a framework for successful environmental enrichment practices. In this paper, we outline the key individuals and events that shaped two of the cornerstones of the modern zoo: (1) the emergence of reward-based husbandry training practices, and (2) the engineering of environmental enrichment. In addition, we (3) suggest ways in which behavior analysis can continue to advance zoo welfare by (i) expanding the efficacy of environmental enrichment, (ii) using within-subject methodology, and (iii) improving animal-visitor interactions. Our goal is to provide a historical and contextual reference for future efforts to improve the well-being of zoo animals.

**Keywords:** animal training; animal welfare; behavior analysis; behavioral engineering; environmental enrichment; operant conditioning; zoos





## 1. Introduction

Due to the many advancements in zoo animal welfare and management, those familiar with modern, accredited zoos might expect to find diverse, enriched exhibits focused on the needs of each species, knowledgeable zookeepers well-versed in animal welfare and training, and visitor education experiences with an emphasis on the conservation of the zoo animals. What is not as apparent is the important role behavior analysis, or the Skinnerian operant conditioning-focused (e.g., reward- or reinforcement-based) approach to behavioral psychology, played in the formation of the present-day zoo. The modern zoo itself can be defined by two major behavioral advances focused on improving welfare: (1) the use of animal training procedures to increase voluntary participation in husbandry or other veterinary procedures by the zoo animals [1–3], and (2) the implementation of environmental enrichment to decrease detrimental and increase species-typical behaviors [4–6]. These advances were developed through decades of research and practices that incorporated behavioral principles to identify desirable outcomes for animals and visitors alike.

For those working with or in zoos, the influence of operant conditioning on animal training may be obvious. In most cases, trainers or keepers deliver reinforcing consequences to modify an animal's behavior, often for husbandry purposes [7,8]. While the connection between behavior analysis and environmental enrichment may be less apparent, the practice of environmental enrichment also began as a set of consequence-focused operant conditioning procedures in zoos. Early enrichment practices incorporated food

delivered mechanically as a reinforcer for engaging in desired responses, such as primates swinging from parts of their exhibit or felids chasing and catching artificial prey [9–11]. These procedures often incorporated visual or auditory stimuli that were meant to elicit or set the occasion for those desired responses, thus functioning as conditional (respondent) or discriminative (operant) stimuli, respectively. The advent of environmental enrichment was therefore a behavioral engineering endeavor, meant to adjust environment-behavior contingencies in the most optimal manner. From these modest beginnings, the practice grew to its current use as the major tool to increase naturalistic behaviors of all the animals we see in the zoo, as well as now extended to other settings, like shelters, homes, and farms [12–15].

The following paper examines the influence of behavior analysis on the modern zoo. We do this in three parts, with the first two parts detailing (1) the emergence of reward-based zoo husbandry training practices, and (2) the engineering of environmental enrichment in zoos. Both will sufficiently detail how behavioral principles were involved, with several photos to provide some context. For the final point, we discuss (3) the future of behavior analysis in zoos, with attention to how behavior analysis can continue to improve the lives of zoos animals by (i) expanding the efficacy of environmental enrichment, (ii) using within-subject methodology, and (iii) improving animal-visitor interactions. Our goal is to therefore detail the important role behavior analysis has had in the formation of the modern zoo, as well as how such behavioral principles can guide continued welfare progress in zoos and similar settings.

## 2. The Emergence of Reward-Based Zoo Husbandry Training Practices

Contemporary animal training procedures are often associated with clicker training or similar uses of conditioned reinforcers paired with positive reinforcement [16–18]. These procedures and other behavior analytic principles to train animals are tied to two major events: (1) Skinner's discovery of shaping, or the use of differentially reinforcing successive approximations to a target response [19–21], and (2) the creation of a field of Applied Animal Psychology by Keller and Marian Breland, two of Professor Skinner's graduate students at the University of Minnesota [22,23]. Both events were also directly connected to Project Pelican (also known as "Project Pigeon"), a wartime effort in the early 1940s that involved training pigeons (*Columba livia*) to guide bombs [24–27]. The project was sponsored by government contracts and through General Mills, Inc., with much of the research conducted within the top floor of the General Mills flour mill building (see Figure 1) in downtown Minneapolis, Minnesota [19,26].

Until Project Pelican, most of Skinner's research involved the use of laboratory rats (*Rattus norvegicus*), with all the research published in The Behavior of Organisms [28] using rats as subjects. It was through Project Pelican that Skinner and his colleagues first began examining behavioral principles with pigeons, as well as his first experience training animals outside of an operant chamber, or "by hand" [24,29], for a review, see [19]. These events resulted in the discovery of shaping, which was so profound that by 1943, Keller and Marian Breland, having worked with Dr. Skinner as University of Minnesota Psychology graduate students on Project Pelican, left academia and began Animal Behavior Enterprises (ABE), an organization dedicated to the training of animals for a variety of applied, profit-driven purposes, including commercials and coin-operated acts [30–34].

By the 1950s, the Brelands moved their business to Hot Springs, Arkansas, where they held a tourist attraction known as "IQ Zoo", intended as both an entertaining and educational experience [25,35]. The Brelands also continued to train animals for coin-operated acts and other revenue-generating ventures (see Figure 2), as well as suggesting the use of naturalistic exhibits and visitor-focused learning opportunities within zoos and similar settings [36–39].

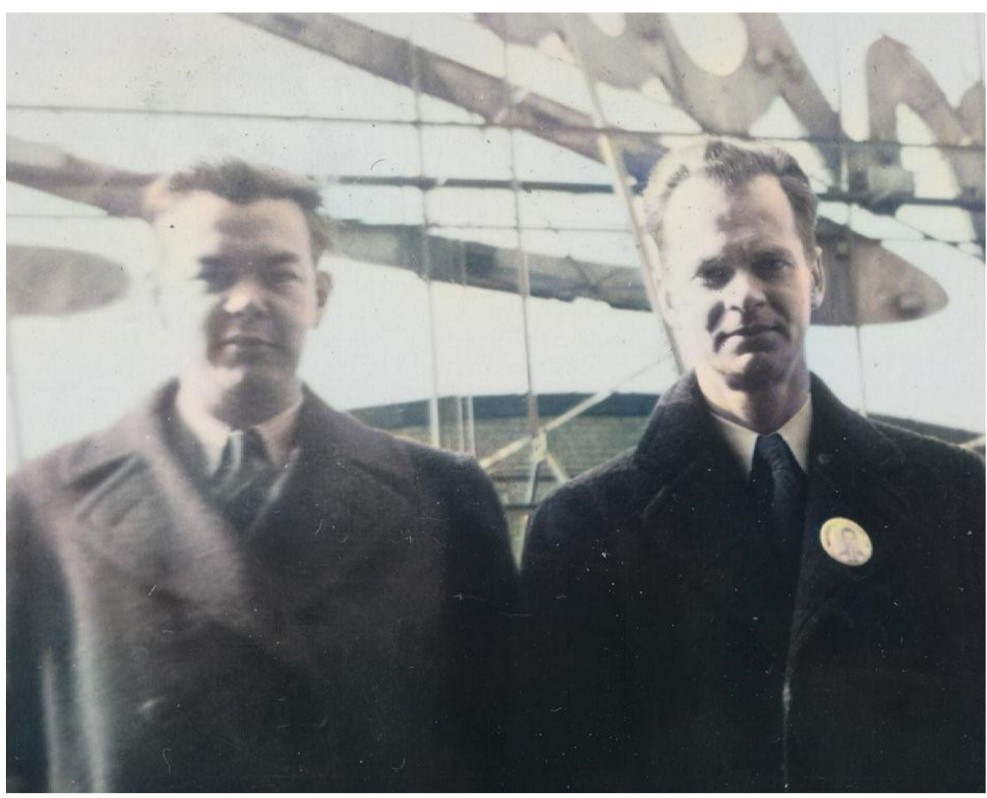

**Figure 1.** Colorized photo of Keller Breland (left) and B. F. Skinner on top of the General Mills building in Minneapolis, MN, USA, circa 1943. (Photo courtesy of Robert E. Bailey).

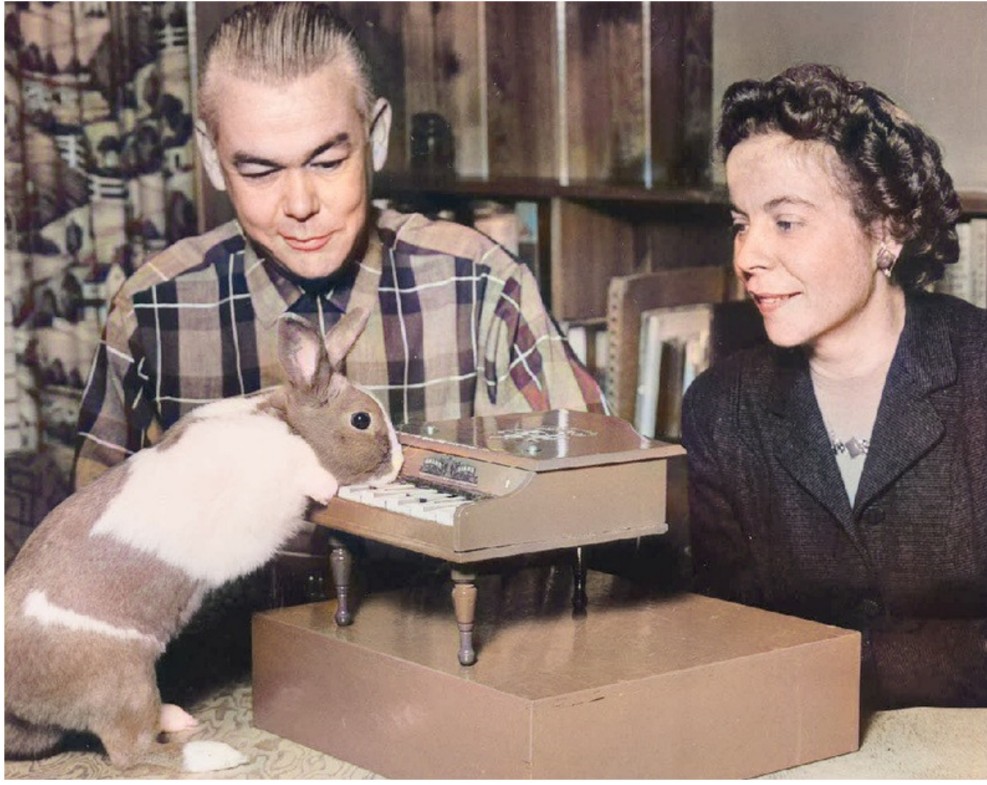

**Figure 2.** Colorized photo of Keller and Marian Breland training IQ Zoo's Professor Punch, late 1950s. (Photo courtesy of Robert E. Bailey).

By 1955, the Brelands also began working with marine mammal parks to establish some of the first cetacean training shows [22,40]. Originally begun with the training of bottlenose dolphins (*Tursiops truncates)* at Marine Studios (now Marineland of Florida, see Figure 3), the use of operant conditioning procedures to train marine mammals would soon spread to other parks, including Marineland of the Pacific, Sea Life Park, and SeaWorld (for a review, see [40]).

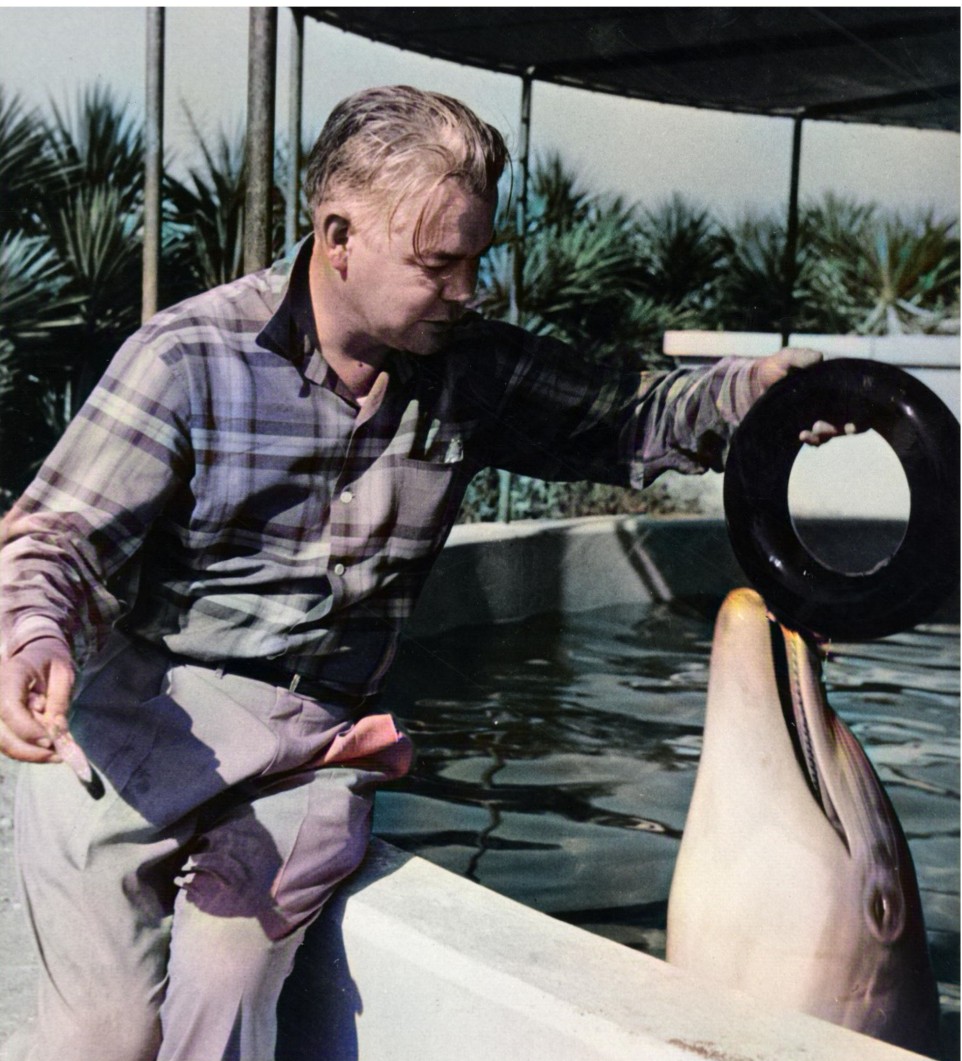

**Figure 3.** Colorized photo of Keller Breland training a dolphin at Marine Studios/Marineland of Florida, circa 1957. (Reproduced from [40] with permission of John Wiley and Sons, 2014).

Eventually, the success of operant conditioning to shape the behaviors of marine mammals would be popularized by books such as Karen Pryor's Lads before the Wind [41] and Don't Shoot the Dog! [17]. At the same time, zoos began to see the benefits of using such procedures to produce voluntary participation in veterinary husbandry practices [7,42]. For instance, San Diego Zoo implemented a shaping protocol that allowed a diabetic drill (*Mandrillus leucophaeus*) to choose to receive insulin injections [43]. Denver Zoo trained nyala (*Tragelaphus angasi)* and bongo (*Tragelaphus eurycerus*) to willingly enter crates to receive blood draws or other veterinary procedures [44,45]. Bloomsmith, Stone, and Laule [46] successfully used reward-based methods to train large groups of chimpanzees (*Pan troglodytes)* to elect to move (i.e., "shift") from outdoor areas to an indoor portion of their enclosures. The use of reinforcement-based training procedures is now commonplace

for many species within most accredited zoos, with some organizations requiring facilities to establish and use such protocols to receive accreditation [47–50].

## 3. The Engineering of Environmental Enrichment in Zoos

The implementation of environmental enrichment in zoos can be traced to Hal Markowitz, who served as Director of the Oregon Zoological Research Center, Associate Director of the Portland/Washington Park Zoo (now the Oregon Zoo), and Professor of Biological Science at San Francisco State University. While prior work in zoos and similar settings described the need for promoting the well-being of captive animals [51–54], Markowitz and his colleagues were among the first to promote a systematic, functional approach to the behavior of zoo animals through behavioral engineering [9,55–59]. The term "behavioral engineering" itself was taken directly from the application of Skinner's operant conditioning procedures, or the field of Applied Behavior Analysis (ABA) [24,60,61]. The term, "engineer," emphasizes the real-world application of a science. In the same way that mechanical engineers apply basic physics principles to better society, behavioral engineers apply the science of respondent and operant conditioning to bring about positive change in the world and, in this case, zoos. Through the creation of contrived, reinforcement-based learning contingencies, Markowitz and his colleagues were able to produce mechanical levers that would allow white-handed gibbons (*Hylobates lar*) to swing across their enclosure to activate the levers and receive a food reward, mandrills (*Mandrillus sphinx*) to compete against zoo visitors in a computerized arcade-like reaction game, and polar bears (*Ursus maritimus*) to vocalize into a voice-operated relay system that would result in a frozen fish being launched into their exhibit (see Figure 4) [62–64]. All the above was carried out to produce desired behaviors (e.g., foraging) or reduce undesired responses (e.g., pacing) as a form of artificial, mechanized occupational therapy for the zoo animals.

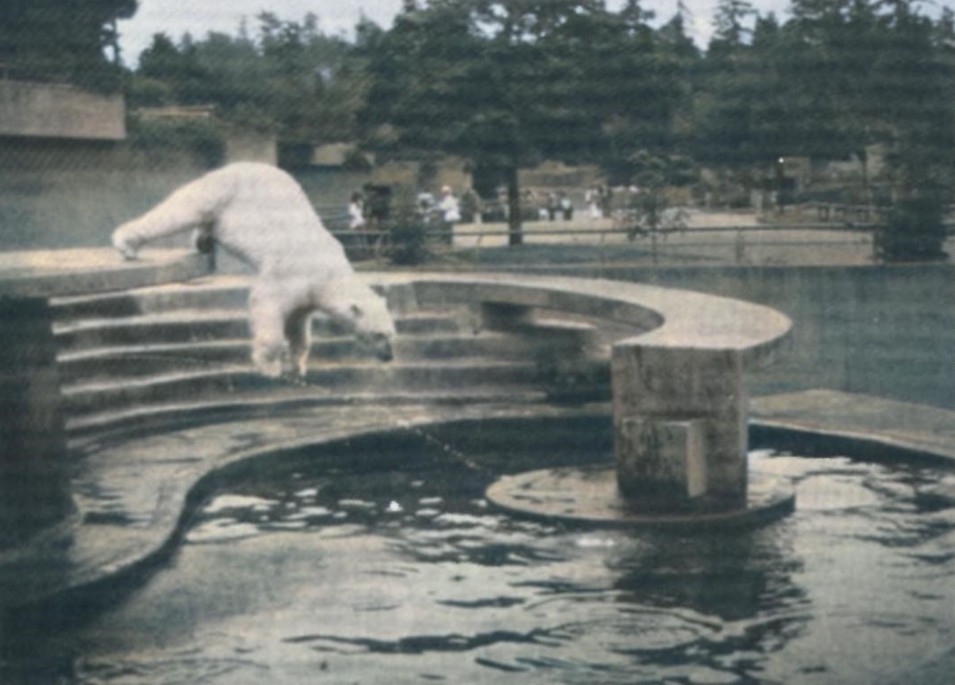

**Figure 4.** Colorized photo of a polar bear chasing a frozen fish catapulted into their pool at the Portland (Oregon) Zoo, circa mid-1970s. (Reproduced from [63] with permission of American Veterinary Medical Association, 1977).

Among the criticisms of such applications were the artificiality of the procedures involved, as well as the arbitrary distinction of what constituted 'desired' responses to be increased [65–67]. These critics argued that, rather than engineering environments through contrived contingencies, zoos should focus on creating naturalistic exhibits that increased

spatial and temporal complexity, for instance, the exhibit arrangement or timing of feeding events [68,69]. Markowitz [70] responded to some of these criticisms by noting that, "the best interests of captive animals may not be served by making their state as 'wild' as possible." (p. 12). Markowitz's argument was that artificial and mechanical contingencies, such as those provided by enabling an elephant to pull a chain to receive part of their diet (see Figure 5), could improve the lives of exhibited animals. "Behavioral enrichment", a term Markowitz began using synonymously with behavioral engineering, could benefit exhibited animals by giving the animals 'something to do' [10,70].

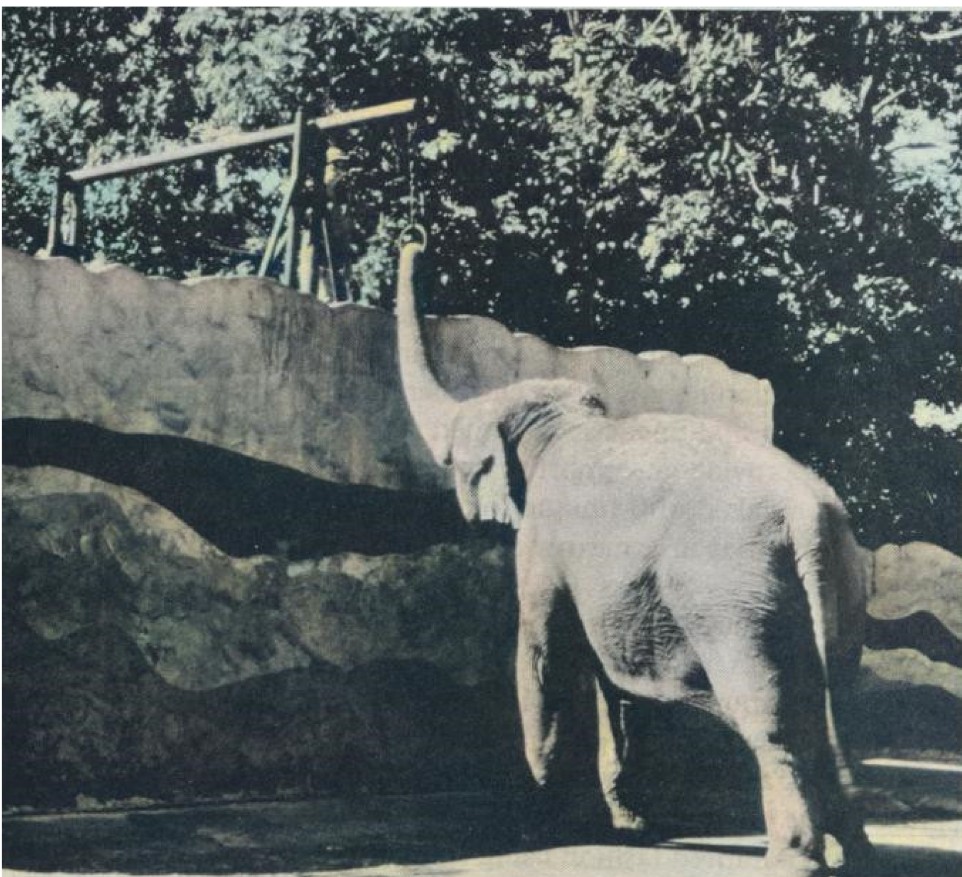

**Figure 5.** Colorized photo of an Asian elephant pulling a ring to obtain fruit at the Honolulu Zoo, circa late-1970s. (Reproduced from [10] with permission of John Wiley & Sons Limited, 1982).

The term "enrichment" appeared to come directly from psychobiological and developmental neuroscience research, where comparisons were often made between animals raised in enriched versus impoverished environments (for a review, see [71]). Regardless, both the ideas of enriching and engineering environments were now being used interchangeably, with other authors arguing that these and the naturalistic/complexity concepts could be integrated to benefit the behavioral welfare of zoo animals [72]. Likewise, some of Markowitz and colleagues' later efforts focused on naturalistic implementations of behavioral engineering endeavors, such as Asian small-clawed otters (*Aonyx cinereus*) hunting live crickets (*Acheta domesticus*) that visitors mechanically assisted in releasing into different parts of their exhibit, servals (*Leptailurus serval*) chasing artificial prey run through clear tubes in their enclosure (see Figure 6), and an African leopard (*Panthera pardus*) chasing bird sounds along a tree limb to receive a food reward [73–75].

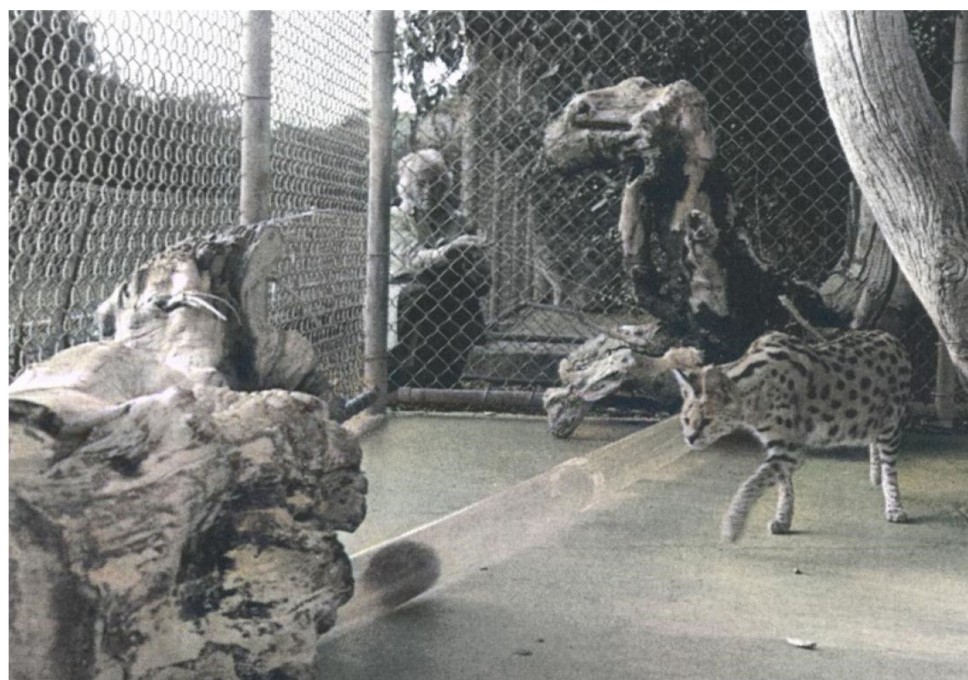

**Figure 6.** Colorized photo of a serval chasing artificial prey at the San Francisco Zoo, circa 1984. (Reproduced from [74] with permission of Elsevier, 1982).

The result has been the introduction of environmental enrichment as an approach that is both pragmatically focused and attends to the species-typical needs of the organisms involved. The use of enrichment is one of those rare events that requires simultaneous attention to both evolutionary and learning histories to be optimally implemented, which necessarily requires an integration between naturalistic and engineered exhibits [14,76]. Equally important for the zoo is how the visitor behaves in response to enriched animals, therefore driving the need for naturalistic devices and responses for and from the animals being enriched, respectively [77,78]. Environmental enrichment, because of its behavior analytic underpinnings, is now an animal welfare endeavor where all features of how an animal interacts with its environment are examined for their behavioral benefits [5,79].

## 4. The Future of Behavior Analysis in Zoos

Behavior analysis has had a profound influence on shaping the modern zoo; however, we have only begun to realize its full potential in animal settings. There have been many calls for the adoption of a behavior analytic framework to improve animal care [7,61,80–86]. While the science of behavior analysis grew out of basic animal studies, recent advances in applied behavior analysis have been developed and implemented primarily in human clinical settings. However, coming full circle, researchers are now successfully adapting and using behavioral protocols developed for use with people to impact animal welfare. For example, functional analysis protocols have been used to assess and treat problem behaviors in animals [87–96], and this function-based approach that emphasizes the identification and modification of existing behavior-environment relationships may help provide a framework that allows animal caregivers both a deeper understanding of behavior and the ability to move beyond the reliance of artificial reinforcers when modifying behavior [61]. In addition, empirical preference assessments have been successfully used in a variety of species [97–111] with promise for improving training effectiveness. By using a behavior analytic lens, adapting existing behavioral technologies, and developing new, animal-specific, behavioral protocols and methodologies, behavior analysts could play a considerable role in guiding the next advances in modern zoos. We outline just a few of the possibilities below, including (i) expanding the efficacy of environmental enrichment, (ii) using within-subject methodology, and (iii) improving animal-visitor interactions.

Applied behavior analysts have a long history with increasing behavioral repertoires to the benefit of the participants involved, and zoo animals should be no exception to this approach. Using both antecedent and consequence manipulations, behavior analysts in clinical settings have developed effective protocols to increase social and occupational skills with people [112,113]. While zoos use environmental enrichment to increase an animal's behavioral repertoire, its application is often void of an underlying theoretical framework and is based on factors like appearance, novelty, cost, or availability/convenience [14,76]. A zoo-based behavior analyst could increase the effectiveness of these enrichment practices by learning theory-focused implementations that incorporate factors such as schedules of reinforcement, habituation, preference, variation, and choice (e.g., [86,109,111,114–116]). In doing so, they could help to solve issues such as animals who are unusually inactive, not using existing enrichment options, or not utilizing all areas of a habitat. Furthermore, training practices could be combined with environmental enrichment (e.g., [1]) to shape more complex behaviors that allow for increased engagement with enrichment devices. Enrichment could also play a key role in training adaptive behaviors that would aid in conservation and re-release programs (e.g., [117,118]). In addition to increasing an animal's behavioral repertoire, another goal of enrichment is to decrease maladaptive behaviors, and behavior analysis likewise can provide guidance in this area. The competing stimuli framework is used in ABA to identify which items or activities effectively reduce problem behaviors by offering alternative sources of reinforcement [118,119]. By monitoring an individual animal's (or group's) enrichment use and problem behavior across different enrichment conditions using within-subjects designs, behavior analysts could make data-driven decisions regarding the best implementation of a variety of different types of potential enrichment items or events [75,80,82,117,120,121].

Within- or single-subject research designs such as some of the research mentioned above form the methodological foundation of behavior analysis [122–124]. These individual-focused designs allow researchers to experimentally determine the functional relationship between variables and effectively monitor a subject's response to interventions using as few as one subject. In ABA, the goal is to make a meaningful change in the behavior of a particular individual in a specific circumstance or setting, and within-subject designs allow for this flexibility and specificity [123]. Similarly, every attempt to provide proper behavioral welfare for any zoo animal is ultimately a study with a sample size ($n$) = 1. Even if it is possible to find enough similar subjects to conduct a robust group study, knowing that the average animal responds to a particular reinforcer or treatment is of limited use when focusing on the treatment of an individual animal. For example, knowing that 60% of a sample of a particular species will forage for a particular food item will not aide a facility if the animal in an exhibit is in the 40% who will not. Animals' responses to stimuli are based on both their species and individual histories as well as their current environment, so assessing and monitoring behavior at the level of the individual level is key. Behavior analytic methodologies and their focus on the overt behaviors of individuals is ideally suited for improving the lives of zoo animals [80,82,125].

Finally, zoos have become increasingly interested in understanding both direct and indirect animal-visitor interactions, with particular emphasis in minimizing adverse impacts that visitors may have on animals while also increasing visitor education and entertainment [77,126,127]. Factors that increase visitor engagement with conservation efforts are also of interest (see [128] for review). Zoo visitors respond favorably to animal interactions, training demonstrations, and environmental enrichment activities [128], and a zoo-based behavior analyst could optimize these events. For example, more interactive elements in the spirit of some of Markowitz's early behavioral engineering endeavors (see 'The Engineering of Environment Enrichment in Zoos' section) could be developed and upgraded with modern technology (e.g., [129–132]) so that visitors could introduce interactive elements (e.g., movement, sound, or food) into animal exhibits. This would require careful arrangements and monitoring, of course, and behavior analysts have the expertise and tools to ensure its success. A large component of understanding these interactions should

come from simultaneously observing the overt behaviors of both animals and visitors, an exercise all too common for many applied behavior analysts when equally working with caregivers and clients. While zoos have a primary responsibility to promote the welfare of the animals in their care, by focusing on learning contingencies involved for both the animals and the visitors, ABA practitioners in zoos could effectively provide win-win solutions that mutually improve the welfare of animals and the education/enjoyment of the visitor.

## 5. Conclusions

Many of the behavioral practices found in modern zoos can be traced back to pioneers in operant conditioning such as B. F. Skinner, Keller Breland, Marian Breland Bailey, and Hal Markowitz. Behavior analytic-driven advances in animal training and environmental enrichment have improved the welfare of zoo animals and have benefited animal care workers and zoo visitors. Nonetheless, behavior analysis still has untapped potential in this setting. While some operant-based techniques, such as clicker training and environmental enrichment, have become commonplace in zoos, these practices have in some instances become disconnected from their underlying learning principles. Further advancement of behavior analysis in animal settings necessitates individuals who are well-trained in the fundamentals of respondent and operant conditioning and can combine this theoretical background with practical knowledge of animal behavior to design habitats and arrange behavioral contingencies to optimize welfare [61]. This will require more animal care professionals with advanced training in behavior analysis and more collaborations between zoos and behavior analysts working in other settings, such as human clinical settings or academic institutions [82,84,85,133]. Integrative approaches to behavior analysis and animal behavior are rapidly increasing. We hope that this historical recognition of such work in zoos, as well as a potential guide for future research, helps foster such practices.

**Author Contributions:** Conceptualization, E.J.F.; writing—original draft preparation, E.J.F., A.L.M.; writing—review and editing, E.J.F., A.L.M. Both authors have read and agreed to the published version of the manuscript.

**Funding:** This research received no external funding.

**Institutional Review Board Statement:** Not applicable.

**Data Availability Statement:** Not applicable.

**Conflicts of Interest:** The authors declare no conflict of interest.

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
