# Peer review of "Animal Training, Environmental Enrichment, and Animal Welfare: A History of Behavior Analysis in Zoos"

_2673-5636, doi:10.3390/jzbg2040038_

Round 1

Reviewer 1 Report

This is a nicely written and comprehensive review about the history of behaviour analysis in zoological parks. As well as being interesting from a historical perspective, it will also be a very useful resource for zoo staff, or other scientists that work at zoos, in order to understand more about effective training, and functionally relevant uses of enrichment.

I have one suggestion that I think would be very useful, if the authors feel there is scope. As the authors will know, once of the roles of zoos in the modern era is to be involved in ex-situ conservation (eg captive breeding and re-introduction of endangered species). I wonder what the authors think that the role of enrichment and training could be in promoting successful re-introduction? In addition, what might the challenges and opportunities be for researchers/conservationists looking to develop effective protocols to maximise the potential for successful re-introduction? It seems to me that there is much potential here, and it is not really mentioned.

Author Response

Thank you, and please see attached our response.

Reviewer 2 Report

Overall it is good to see a paper with this scope and topic.  I feel it might benefit from being more explicit on the implications of this framing in a few areas, with the zoo professional audience in mind.  Overall it is phrased and presented in a passive way that is not especially useful or inspiring to the readership of this journal.

 [L45] Enrichment is, these days, more often linked to the idea of species-specific natural behaviors and/or “novelty”.  This historical motivation of providing featured with functions that are rewarding (artificially or innately) has been largely--but not entirely-- lost for many people in the field.  I would agree that reasserting the idea of enrichment as the support of functional/rewarded behavior would be beneficial in a number of ways including identification of pointless/aesthetic approaches to enrichment and balancing the role of natural versus artificial mechanisms of improving animal wellbeing. (It would also help clarify how to properly measure efficacy and compare food rewarded versus otherwise motivated activities. Both of these being enduring problematic issues for enrichment researchers.) All of this needs to be stated very clearly without presupposing familiarity with the norms of behaviorist thinking which is getting rather rare in zoo professionals of pre-retirement ages.

It is great to revisit [L149] the natural versus artificial argument and it reminds me that this discussion continues to happen as if it is a brand new idea. Again I feel that you provide a solution in that measuring the functionality of the behavior with complete contingencies and natural form without the development of maladaptive behaviors/injury/conflict is the best way to compare and choose the most idea enrichments—rather than predetermining which is best based on what humans see as “natural”.  (That is, start with giving them “something to do” [L60] and graduate to giving them “something good to do” based on how it affects overall wellness (positive emoting, choice-contrafreeloading, fitness, etc). This is not a remedial thing to assess but also not impossible.  It takes more than a few sentences, and preferably a modern example or two, for people to move their thinking into a whole new paradigm. This might also be helped by a photograph of said example in a modern zoo.

I love the part at L179 but feel much more is needed here for the paper to specifically address the issue of biological preparedness and innate behaviors versus those learned by an operant or respondent mechanism.  Currently there is often an artificial dichotomy in how these are considered which also feeds into fallacies such as that all species-specific behaviors are good or all stressful behaviors are bad. People seem to either strive for one simple way to assess interventions or give up on assessing them all together and taking the “kitchen sink” approach.  Markowitz for all his genius blew off the idea that more naturalistic setups might tend to be demonstrably superior. A possibility that remains, as far as I know, untested to this day.

[Re L70] I think many people would benefit from revisiting the historical roots of these subjects.  But perhaps the authors could also be more explicit in why these approaches persist more in some areas than others and why these roots are increasingly obscured. This would assist in demonstrating how reconnecting to more reductionist principles and mechanisms will improve and reinvigorate research and practice without negating the use of more holistic ideas like welfare, stress or biological fitness.

L164--It might be worth visiting how this origin of enrichment in early life studies contributed to conflating enrichment with complexity and from there with novelty.  Such that some now consider only novelty a form of enrichment.  Thinking about what rewards the behavior helps unpack this problem and direct people towards enrichments not supported only and entirely by motivations to explore or investigate.

L208 the following is likely to be opaque to zoo professionals, perhaps give examples “Applied behavior analysts have a long history with increasing behavioral repertoires  to the benefit of the participants involved”

L230 The following could also be better unpacked for  readers not already familiar with the behaviorist paradigm “knowing that the average animal responds to a particular reinforcer or treatment is of limited use if the 231 animal in question does not respond in the average way.”

L239 I think you really need to be explicit that one other big area where behaviorist approaches persists is working with people needing substantial caregiver support due to disability (re “an exercise all too common for many applied behavior analysts when equally working with care-240 givers and clients.) … As well as school and prisons of course where the issues of coercion and benefit to the client are also highly relevant and not always well achieved.  The history of ABA being entangled with abuse might need some discussion to explicate how this is avoided. It contributes to why these approaches fell from grace. BA remains the most useful to help individuals who are captive or cognitively limited, but for both good and bad reasons. 1) it works, 2) it does not have to deal with clients rejecting our outdated and objectifying language and sometimes practices.  Leading to 3) ever BA plan needs to connect to holistic paradigms and outside perspectives for third party advocacy, validation of real benefit, and ethical review. Something that, usefully for this audience, validates also having people learn about things other than BA.

L243 I would argue win/win solutions are great when present, but this outcome will not uniformly be available, and certainly not always optimal.  Any factor that is independent will have a range of interactions even when approached very inventively.  However if the authors have ideas about how animals with zero interest in humans might benefit from them, a brief example would be good to share. I would still say that ethically there always has to be a primary (non-equal) interest and here is to benefit the animal or in some cases species conservation.

L253 I suspect I am not the only one with a reaction to the old “shoulders of giants” phrase that is a bit “okay boomer” (we all have out triggers). It feels like being told to just go back and do it like they did then, because little of worth has happened since.  A frame that is more reintegrative will probably be more welcome to this readership. When you compare the overall functionality of zoos between in the 1950s and now they have benefited from their own giants in habitat design, nutrition, genetics, ethology, veterinary care, reproduction, conservation advocacy, humane education etc etc. (Some things have not improved or even got worse, but overall, huge progress).  IMHO the two giants need to marry and you might be encouraging them to fight.

Essentially, I commend this explication of the Skinnerian basis of Zoo-based training and enrichment,  But the zoo field needs to have a vision for these principles that is not mired in images of circus acts and the language of control (engineering, modification) which younger generations are not very interested in embracing.  That is, BA is informed by contextual information to be 1) minimally coercive, and 2) maximally likely to benefit the animal’s interests. Given that Skinner was essential a Utopian think this is not an arduous task, mainly a matter of adjusting some modernist language and actively engaging with current day intersectional thinking.

The “true behavioral engineers” you hope will come are not going to use the words “true” or “engineer” and will probably be better for it. Those in zoos do need to know more behavior analysis, but also those in behavior analysis need to get the hell back out into the field to learn what is required there-- at all the levels of behavior selection including the verbal community of the users, funders and audience of the zoological park. This requires getting real about how BA threw itself under the bus and now barely any universities in America have a psychology department with any interest in animal applications such as zoos—and where they do they have been gazzumped by other approaches (e.g. pet cognition).  If zoos are meant to go to BA, BA needs to find a way to go to zoos.

Author Response

(The authors gave the same response as above.)

Round 2

Reviewer 2 Report

While I feel the overall tone is still old-fashioned and not really phrased in a way to connect with the audience of this journal, it nevertheless makes important points that they should benefit from